# Participation of Somatic Stem Cells, Labeled by a Unique Antibody (A3) Recognizing Both *N*-glycan and Peptide, to Hair Follicle Cycle and Cutaneous Wound Healing in Rats

**DOI:** 10.3390/ijms21113806

**Published:** 2020-05-27

**Authors:** Chisa Katou-Ichikawa, Hironobu Nishina, Miyuu Tanaka, Shigeo Takenaka, Takeshi Izawa, Mitsuru Kuwamura, Jyoji Yamate

**Affiliations:** 1Laboratory of Veterinary Pathology, Osaka Prefecture University, Izumisano City, Osaka 598-0048, Japan; chai.grn@gmail.com (C.K.-I.); nishina.hironobu@ma.mt-pharma.co.jp (H.N.); cmt21892@osakafu-u.ac.jp (M.T.); izawa@vet.osakafu-u.ac.jp (T.I.); kuwamura@vet.osakafu-u.ac.jp (M.K.); 2Department of Clinical Nutrition, Osaka Prefecture University, Habikino City, Osaka 583-8555, Japan; takenaka@rehab.osakafu-u.ac.jp

**Keywords:** antibody, cutaneous wound healing, hair follicle cycle, *N*-glycan, somatic stem cells

## Abstract

A monoclonal antibody (A3) was generated by using rat malignant fibrous histiocytoma (MFH) cells as the antigen. Generally, MFH is considered to be a sarcoma derived from undifferentiated mesenchymal cells. Molecular biological analyses using the lysate of rat MFH cells revealed that A3 is a conformation specific antibody recognizing both *N*-glycan and peptide. A3-labeled cells in bone marrow were regarded as somatic stem cells, because the cells partly coexpressed CD90 and CD105 (both immature mesenchymal markers). In the hair follicle cycle, particularly the anagen, the immature epithelial cells (suprabasal cells) near the bulge and some immature mesenchymal cells in the disassembling dermal papilla and regenerating connective tissue sheath/hair papilla reacted to A3. In the cutaneous wound-healing process, A3-labeled epithelial cells participated in re-epithelialization in the wound bed, and apparently, the labeled cells were derived from the hair bulge; in addition, A3-labeled immature mesenchymal cells in the connective tissue sheath of hair follicles at the wound edge showed the expansion of the A3 immunolabeling. A3-labeled immature epithelial and mesenchymal cells contributed to morphogenesis in the hair cycle and tissue repair after a cutaneous wound. A3 could become a unique antibody to identify somatic stem cells capable of differentiating both epithelial and mesenchymal cells in rat tissues.

## 1. Introduction

Monoclonal antibody is an indispensable tool for biological science, as well as the medical field, for regenerative therapy. If such antibody has high specific antigen capable of recognizing a certain epitope that may regulate cellular functions such as cell differentiation, survival and death, immunohistochemistry with the antibody is useful to identify cells expressing the epitope [1]. Some antibodies recognizing the cluster of the differentiation (CD) 34, CD90 and stage-specific-embryonic antigen (SSEA) have been used for identification of stem cells, because epitopes are expressed in immature cells in the body [2]. These antibodies should be useful for studies on the stem cell niche.

We developed a unique monoclonal antibody (named A3); A3 was generated by using rat malignant fibrous histiocytoma (MFH)-derived cultured cells as the antigen [3]. Based on the gene expression profiling, functional analysis and histopathological findings of MFHs, it has been considered that MFH may be derived from mesenchymal stem cells or undifferentiated mesenchymal cells; therefore, human MFH is also called pleomorphic undifferentiated sarcoma [4].

Interestingly, in addition to rat MFH-constituting cells, A3 could label immature mesenchymal cells among visceral organs in rat fetuses [5]. In adult rats, furthermore, vascular pericytes and bone marrow-constituting cells were also labeled with A3 immunohistochemistry; the pericytes and cells in the bone marrow are considered to be immature mesenchymal cells, although the cellular nature should be investigated further [6,7]. More interestingly, it was found in rat fetuses and neonates that A3 labeled epithelial cells in the hair germ and peg in developing hair follicles, as well as epithelial cells in the outer root sheath adjacent to the bulge in mature hair follicles; the A3-positive epithelial cells are regarded as suprabasal immature cells in the developing epidermic hair follicle. Additionally, spindle-shaped mesenchymal cells surrounding the hair peg and mature hair follicle reacted to A3 [8]. A3-reacting cells in the developing rat fair follicles may be stem cells with the potential to differentiate into either epithelial or mesenchymal cells.

Collectively, A3 is regarded as an antibody recognizing somatic stem cells in rat tissues [5,8]. However, epitopes recognized by A3 remain to be investigated. It has been reported that stem cells in the bulge in hair follicles or epidermal progenitors such as suprabasal cells may contribute to hair cycling and cutaneous wound repair [9,10,11]. In addition, immature mesenchymal cells in the connective tissue sheath of hair follicles could participate in the wound-healing process [12]. In this study, we analyzed the molecular biological features of the epitope recognized by A3 and then investigated the possible participation of somatic stem cells labeled with A3 immunohistochemistry in the hair follicle cycle and cutaneous wound repair (epidermal regeneration) in rats. It was found that A3 could be a useful marker antibody that recognizes *N*-glycan and the amino acid sequence in rat somatic stem cells.

## 2. Results

### 2.1. Molecular Biological Analysis of A3-Recognizing Antigen

#### 2.1.1. The Characteristic of A3-Recognizing Antigen on MT-9 Cells

MT-9 cells were polyhedral and spindle in shape. A3-signals were detected diffusely on the surface of MT-9 cells and as fine granules in the cytoplasm (Figure 1A).

Western blotting was performed to identify the molecular weight of A3-recognizing antigen. The A3-signal band was strongly detected in the sample prepared under nondenaturing conditions with a wide range of molecular weights between 75 to 100 kDa (Figure 1B). Periodic acid-Schiff (PAS) reagent reacted mostly to the same band with A3. The A3-signal band was faint in the boiled sample, and the corresponding bands disappeared in the sample prepared under denaturing conditions.

To further investigate for the characteristics of the A3-recognizing antigen, a glycosidase digestion experiment was performed. The A3-signal band, which was applied with a glycolytic enzyme, appeared at a lower molecular weight between 50 to 75 kDa as a weaker signal than the signal band of the intact MT-9 lysate at 75 to 100 kDa (Figure 1C). The band disappeared by treatment with glycopeptidase F (GPF).

#### 2.1.2. A3-Labeled Cells in Bone Marrow of Adult Rats

In the bone marrow of adult rats, polyhedral- and spindle-shaped mesenchymal cells were labeled with A3 (Figure 2A). Some A3-labeled cells coexpressed various antigens such as CD90 (Figure 2D–F) and CD105 (Figure 2G–I) as mesenchymal stem cell markers [13], as well as rat endothelial cell antigen-1 (RECA-1) (Figure 2J–L) for rat endothelial cell marker. In addition, in consecutive sections, the localization of A3 was similar partly to cells expressing CD73 for the mesenchymal stem cell marker (Figure 2B,C), whereas the A3-positive signals did not correspond to CD44-expressing cells (Figure 2M–O); CD44 is a mesenchymal stem cell marker.

#### 2.1.3. A3 Immunoexpression in the Hair Cycle

To further investigate the relevance of the A3-labeled cell to the stem cell, we investigated the participation of A3-labeled cells in the hair follicle cycle. In the telogen, A3 labeled a few cells in the permanent region, lower isthmus and infundibulum (Figure 3A). In the early stage of anagen (Figure 3B–E), A3 reacted to both the epithelial and mesenchymal cells; A3-labeled epithelial cells were located in the bulge (Figure 3D) and hair germ (Figure 3D,E), where they were activated to regenerate towards the variable region (inferior region and hair bulb), as well as the isthmus and infundibulum, with a stronger signal intensity (Figure 3B,C) than in the telogen (Figure 3A); A3-labeled mesenchymal cells were located in the disassembling dermal papilla (Figure 5B–E) and regenerating connective tissue sheath (Figure 3D,E). As hair follicle grew, A3-labeled epithelial cells expanded from the bulge region to both the upper side (infundibulum) and lower region (suprabulbar region) along the outer root sheath with strong intensity (Figure 3G). Similarly, A3 reacted to mesenchymal cells in the regenerating connective tissue sheath with a strong signal intensity (Figure 3G); an A3-positive signal in the regenerating hair papilla (Figure 3F) gradually disappeared from the mature papilla in the anagen phase (Figure 3H). In the catagen, the number and intensity of A3-labeled epithelial and mesenchymal cells gradually declined as the hair follicle regressed (Figure 3I,J); the epithelial cells, which were attaching on the hair papilla, transiently showed a positive reaction to A3 only in the middle stage of catagen (Figure 3I). In the final stage of catagen, A3 labeled some epithelial cells in the bulge, hair germ, isthmus and infundibulum, as well as the dermal papilla (Figure 3J).

#### 2.1.4. Characteristics of A3-Labeled Cells in the Hair Cycle

To further characterize the cell type in the hair cycle, the double immunofluorescence analysis was conducted. A3-labeled epithelial cells were regarded as suprabasal cells located under the companion layer expressing cytokeratin (CK) 6 (Figure 4A–C) and above the basal layer expressing CK15 (Figure 4D–F). A3-labeled cells were partly colocalized with that of cells reacting to leucine-rich repeat-containing G-protein coupled receptor 6 (Lgr6) (Figure 4G–I). The localization of A3-labeled cells was almost similar to that of cells reacting to CK19 (Figure 4J,K). These findings indicated that A3-labeled epithelial cells are suprabasal cells with immature natures. On the other hand, A3-labeled mesenchymal cells were corresponding partly to cells reacting to CD90 (Figure 5A–C), CD34 (Figure 5D–F) or nestin (Figure 5G–I).

### 2.2. Wound-Healing Process

#### 2.2.1. Histology of the Wound-Healing Process

Next, we investigated the relevance of A3-labeled cells to tissue regeneration: the wound-healing process. Cutaneous wound healing proceeded along the three overlapping pathological stages: inflammation (post wounding (PW) days 1–3) (Figure 6A), proliferative (PW days 3–12) (Figure 6B,C) and remodeling (PW days 12–26) (Figure 6D–F) stages. After the punched wounds, skin tissues including the epidermis to cutaneous muscle were lost. On PW days 1–3 (Figure 6A), in the inflammatory stage, the wound was covered by debris/degenerative cells and reactive cells such as neutrophils, fibrin and hemolyzed red blood cells, forming a crust. The epidermal layer became thin at the edge of the wound. On PW day 5 (Figure 6B), in the proliferative stage, the wound bed was filled with gradually forming granulation tissues composed of myofibroblasts (Figure 6B; inset, α-smooth muscle actin (SMA)-expressing myofibroblasts), inflammatory cells and newly formed blood vessels (neovascularization). The epidermis of the wound edge was gradually thickened, and the epidermal cells at the edge were extending and covering the wound surface, with forming stratified layers, indicating re-epithelization. The hair follicles at the edge of the wound were destroyed on PW days 1–5. However, on PW days 7 and 9 (Figure 6C) at the proliferative stage, the hair follicles at the edge of wound were in the anagen phase, and the thickened epidermis extending from the edge covered the wound surface. The anagen hair follicles at the edge of the wound bed and the thickened epidermis were continuously seen on PW day 15 (Figure 6D). On PW days 20–26 (Figure 6E,F), in the remodeling stage, the hair follicles showed the catagen phase, and epidermal hyperplasia with thickness and invagination into the recovering wound bed was clearly seen; the cellularity in the granulation tissues was the plateau and then decreased gradually.

#### 2.2.2. Distribution of A3-Labeled Cells in Cutaneous Wound Healing

In the wound-healing process, an A3-positive reaction was seen in both the epithelial and mesenchymal components of the hair follicles (Figure 7B,C), as well as the re-epithelialized epidermis continuing to the hair follicle at the wound edge (Figure 7C; inset and D); especially, the signal intensity was upregulated at the late-proliferative-to-early remodeling stages (Figure 7B–D). This signal intensity declined in the late-remodeling stage (Figure 7E,F), although these mesenchymal and epithelial intensities in the hair follicle were still greater than that in the control skin (Figure 7A). These findings indicated that A3 labels both epithelial and mesenchymal cells, contributing to the wound healing; especially, the A3-labeled epithelial cells are important for re-epithelialization at the late-proliferative-to-early remodeling stages of the wound-healing process. The epithelial cells of the re-epithelialization might be immature in nature, because they reacted to A3. Additionally, pericytes around the newly formed blood vessels in the granulation tissue reacted to A3 (Figure 7G–I), as reported previously [14].

#### 2.2.3. Characteristics of A3-Labeled Epithelial Cells

To further investigate the cellular characteristics of A3-labeled epithelial cells, we examined wound-healing sections by using double immunofluorescence for A3 with immature cell markers such as CK15 (Figure 8), CD34 (Figure 9) and nestin (Figure 10A–C), as well as using single immunolabeling for CK19 (Figure 10E,G) and proliferating cell nuclear antigen (PCNA) (Figure 10H,I).

Although the CK15-positive cells were located usually in the basal layer of the hair follicles, CK15-positive cells were also observed above the basal layers (suprabasal layer) recognized by A3, especially in the epidermal invaginations (Figure 8D–F) and hair follicles at the wound edge (Figure 8J–L). Moreover, immunoexpression to A3 was seen in the basal layer, especially in the infundibulum (Figure 8G–I), where the expression was not seen in the control skin (Figure 7A). A3 reacted to some CD34-positive epithelial cells around the bulge (Figure 9A–F), and the A3-labeled cells also showed reactivity to nestin (Figure 10A–C). The CK19- and A3-labeled cells were seen in regenerating epithelial cells: that is, the hair follicle epithelial cells around the bulge (Figure 10D,E) and re-epithelialized epidermis (Figure 10F,G). These findings might indicate the continuity from the bulge to re-epithelialized epithelial cells. Generally, proliferating cell nuclear antigen (PCNA)-positive proliferating hair follicular epithelial cells were seen at a few outer layers of the outer root sheath (Figure 10H). Although some A3-positive suprabasal cells reacted to PCNA, the majority of the A3-positive cells above the PCNA-positive basal layer did not show the proliferation activity (Figure 10I), indicating that A3-positive suprabasal epithelial cells did not always have a high proliferative activity.

#### 2.2.4. Characteristics of A3-Labeled Mesenchymal Cells

To further investigate the characteristics of A3-labeled mesenchymal cells, we examined the wound-healing sections by using double immunofluorescence with immature cell markers such as CD90, CD34 and nestin, as well as α-SMA (for myofibroblasts) and RECA-1 (for endothelial cells) (Figure 9G–I). Some A3-labeled mesenchymal cells reacted simultaneously to CD90 (Figure 11A–E), CD34 (Figure 9A–C,G–I) and nestin (Figure 11F–H). CD90 reactivity was much greater in the connective tissue sheath-forming cells at the wound edge than those outside the wound bed at the late-proliferative and early remodeling stages of the wound-healing process (Figure 11B); interestingly, A3-labeled mesenchymal cells also showed a positive reaction to CD90, with an upregulated intensity like hair follicle development [8] (Figure 11A–E). CD34 also reacted to mesenchymal cells in the connective tissue sheath with upregulated signaling from the late-proliferative-to-early remodeling stages of the wound-healing process and reacted to dermal mesenchymal cells in the surrounding tissues; however, mesenchymal cells in the granulation tissue did not react to CD34 (Figure 9A–C). A3 reacted to some CD34-positive cells in the connective tissue sheath (Figure 9A–C,G–I). Some A3-labeled cells were colocalized with nestin-positive cells (Figure 11F–H), although the signal intensity to nestin did not always show clear change throughout the wound-healing process.

In the wound bed, α-SMA-positive myofibroblasts were diffusely seen as the most common cells, with the maximum in the proliferative stage (Figure 6B; inset), and decreased in the remodeling stage, as reported previously [15,16,17]. In addition, α-SMA was expressed in some mesenchymal cells in the connective tissue sheath of hair follicles (Figure 11I–K). The α-SMA signal intensity in the connective tissue sheath was not changed throughout the wound-healing process. The A3-labeled cells seen in the connective tissue sheath did not coexpress α-SMA (Figure 11I–K), indicating that A3-labeled mesenchymal cells in the connective tissue sheath are different from myofibroblasts reacting to α-SMA.

## 3. Discussion

A3 was generated as a monoclonal antibody using rat cultured MFH cells as the antigen [3]. MT-9 cells established from a rat MFH showed immature mesenchymal nature, and it was reported that MT-9 cells had the capacity to differentiate towards adipocytes, myofibroblasts and osteoblasts under appropriate stimuli [4,5,18]. In this study, A3 strongly labeled MT-9 cells in vitro. The epitope recognized by A3 is located mainly on the cytoplasmic membrane and, occasionally, in the cytoplasm of A3-labelled cells [5].

Carbohydrate- or glycopeptide-recognizing antibodies, such as CD34, CD90 and SSEA antibodies, are widely used as stem cell markers. Carbohydrate is a landmark for differentiation. In this study, the A3-recognizing antigen was prepared from MT-9 cells under a nondenaturing condition. The denaturing procedure, such as boiling and adding reducing reagents, declined the reactivity of A3, and the A3-recognizing antigen had a carbohydrate detectable by PAS stain. The reactivity of A3 to the antigen was declined, and the molecular weight was moved to the lower site by digestion with GPF, which has specific a cleavage of *N*-glycans (GlcNAc-Asn bounds) in glycoproteins. These findings indicate that A3 is a conformation-specific antibody that recognizes both *N*-glycan and peptides.

The bone marrow contains stem cells, such as hematopoietic and mesenchymal stem cells [13]. The mesenchymal stem cells may show multipotentials toward mesenchymal and epithelial elements [19]. In this study, A3-labeled cells in rat bone marrow had immature mesenchymal natures, because some of these cells reacted simultaneously to antibodies against CD90 and CD105; CD90 and CD105 antibodies have been used to detect mesenchymal stem cells [13]. On the other hand, the A3-positive signals did not correspond to CD44-expressing cells. The epitope recognized by A3 did not always look the same as that recognized by the CD90, CD105 or CD44 antibodies. The differences in nature of mesenchymal immature cells and epitopes recognized by these antibodies should be investigated further. Interestingly, A3 labeled endothelial cells not only in the bone marrow but, also, in neovascularity such as in granulation tissues in wound healing and in regenerating tissues in colitis [20]. It has been reported that human embryonic stem cells (ES cells) and induced pluripotent stem cells (iPS cells) show direct differentiation to endothelial cells [21,22]. It was considered that A3 could label endothelial cells (neovascularization in regenerating tissue), which might be derived from somatic stem cells in the bone marrow. Collectively, A3-labeled mesenchymal cells seen in the bone marrow are regarded as somatic stem cells [5].

We next investigated the participation of A3-labeled cells in the hair follicle cycle, because hair follicles have stem cells in the bulge with differentiation toward hair follicle-constituting cells. In the telogen phase, A3-labeled epithelial cells were located at the permanent region (lower part of isthmus and infundibulum closed to the bulge). It is reported that the areas contain immature epithelial cells contribute to maintaining interfollicular epidermal homeostasis and the sebaceous glands [10,23]. By contrast, the dermal papilla mesenchymal cells in the telogen phase did not react to A3. These findings correspond to results of the hair development; the dermal condensate at the early stage in rat fetuses did not react to A3 [8]. In the anagen phase, as the hair follicle grew, the signal intensity to A3 was upregulated in both the epithelial and mesenchymal components. It has been reported that hair follicles show epithelial differentiation to the epidermis and hair follicular component along the long axis in the outer root sheath; the immature epithelial cells (suprabasal cells) near the bulge maintain the stemness and dedifferentiation toward the bulge stem cells in the catagen; moreover, these cells regulate the bulge stemness [24,25,26]. Meanwhile, it has been reported that A3-labeled cells around the crypts are regarded as cells participating in the intestinal stem cells niche [27]. Thus, the A3-labeled immature suprabasal cells may contribute to maintaining the hair cycle. Similarly, A3 reacted to some mesenchymal cells in the disassembling dermal papilla and regenerating connective tissue sheath/hair papilla in the anagen. It is shown that undifferentiated mesenchymal cells in the connective tissue sheath can differentiate into mesenchymal cells in the hair papilla [28]. Therefore, A3 could react not only to differentiating mesenchymal cells in the connective tissue sheath but also to regenerating hair papilla. In the catagen phase, the signal intensity to A3 in both the epithelial and mesenchymal components gradually decreased as the hair follicle regressed. However, A3 transiently labeled matrical cells attaching on the hair papilla. The A3-recognizing antigen would appear on the immature cells in the dedifferentiation status of epithelial components attaching on the hair papilla; therefore, these cells in the hair papilla might show plasticity in different differentiations. Moreover, the mesenchymal cells, which were relocated from the connective tissue sheath and hair papilla to the dermal papilla, could be dedifferentiated toward a more primitive status, showing various expressions of the A3-recognizing antigen. These findings in the hair cycle are summarized as follows: A3 could label immature cells of epithelial and mesenchymal components, which might be in the differentiating and dedifferentiating processes, thereby participating in morphogenesis in the hair cycle.

The cutaneous wound-healing process is supported by surrounding tissues, including hair follicles. The hair follicle showing the anagen phase at the edge of the wound bed has been considered to be the cell supplier of the epithelial and mesenchymal cells to maintain the homeostasis when the skin was disrupted; the supply may occur especially at the late-proliferative-to-early remodeling stages of the wound-healing process [29,30]. A3 reactivity was characteristically seen in both the epithelial and mesenchymal cells; the epithelial cells were present at the hair follicles and re-epithelialized epidermis, and the mesenchymal cells were located in the connective tissue sheath at the wound edge. CK15, CD34, CK19 and nestin have been used as immature cell markers in hair follicles in human and mouse skins [26,31,32,33,34]; the CK15 antibody is the most widely used marker to identify the bulge stem cells [33]. CK19-positive cells have been hypothesized to arise from CK15-positive cells in the hair follicle, because the positive signal to CK19 has been reported to be localized above double-positive cells to CK15 and CK19 [26]. The nestin antibody is regarded as a multipotent follicular stem cell marker located in the bulge [35]. The CD34 antibody is used as an immature epithelial cell marker in murine hair follicles; moreover, CD34 can be expressed by different cell populations, depending on the degree of maturation of the hair follicle [26,33]. When the epidermis is disturbed (such as wound), hair follicle-constituting cells contribute to wound regeneration (particularly, re-epithelialization) [29]. In addition, it was reported that A3-positive cells contributed to the tissue regeneration in rat colonic lesions, of which the situation was regarded as a stem cell niche [20]. In colonic lesions, A3-positive cells aggregated beneath the desquamated mucosa, where the re-epithelialization occurred nearby, and the cells reacted simultaneously to CD90, vimentin and CK19 [20]. In this study, the signal intensities to preexisting immature cell markers (as mentioned above) in the hair follicle at the edge of the wound bed were upregulated, as compared with those in the control skin; these positive cells in the present wound healing were corresponding partly to A3-positive cells. These findings indicate that the epithelial cells in the hair follicle in wounded skin would be more activated to re-epithelialize than in normal conditions. Moreover, A3 could react to immature epithelial cells in the epithelial differentiation lineage, because A3-labeled cells existed near epithelial cells reacting to immature cell markers CK15, CK19, CD34 and nestin and coexpressed these antigens in varying degrees.

Antibodies to CD90, CD34 and nestin have been used as immature mesenchymal cell markers [12,23,33,35,36,37,38]. CD90-positive mesenchymal cells in the connective tissue sheath are reported as mesenchymal stem cells having multidifferentiation potentials [39], and CD34-positive immature mesenchymal cells show multilineage differentiations. Mesenchymal nestin-positive cells are regarded as stem cells with neural and mesoderm differentiation potentials. In this study, A3-labeled cells colocalized with some CD90- and CD34-positive mesenchymal cells surrounding the hair follicle at the edge of the wound bed with expanded positive areas. Although CD90 and CD34 recognized different populations of immature mesenchymal cells in the connective tissue sheath, A3 labeled some cells reacting to either CD90 or CD34. In addition, a few A3-labeled cells colocalized with nestin-positive cells. Nestin-positive cells did not change the signal intensity throughout the wound-healing process, indicating that the nestin-positive cells would not always contribute to the wound-healing process, or these cells may have other functions in the connective tissue sheath. A3-labeled mesenchymal cells were clearly different from α-SMA-positive myofibroblasts capable of collagen production in fibrogenesis [17].

Collectively, this study on the wound-healing process showed that A3-labeled epithelial cells participated in re-epithelialization after a cutaneous wound, and the labeled cells may be derived from the bulge cells as immature epithelial cells (suprabasal cells). In addition, A3-labeled immature mesenchymal cells in the connective tissue sheath of the hair follicle at the wound edge showed the expansion of the immunolabeling, indicating the importance of the A3-labeled cells for the regeneration of wounded tissues. A3 recognizes the carbohydrate chain expressed in immature somatic stem cells in rats, as mentioned above; therefore, both epithelial and mesenchymal cells reacting to A3 in the hair follicle could keep the stemness in the wound-healing process. It is also known that the expression level of the carbohydrate chain may be changed during the cellular development from pluripotent and embryonic stem cells [1,2], suggesting that, conversely, *N*-glycan might influence such cellular differentiations as the stemness. The biological properties of epitope recognized by A3 should be investigated further.

## 4. Materials and Methods

### 4.1. Cell Culture

A cultured cloned cell line, MT-9, was used; MT-9 was derived from a spontaneously occurring rat MFH [3]. MT-9 cells have been subcultured in Dulbecco’s modified Eagle’s medium (DMEM) (Thermo Fisher Scientific Inc., Waltham, MA, USA) added with 10% fetal bovine serum (FBS) (Funakoshi Co. Ltd., Tokyo, Japan) in a Nunc EasYFlask 25 cm^2^ or 75 cm^2^ (Thermo Fisher Scientific Inc., Waltham, MA, USA) at 37 °C and 5% CO_2_ atmosphere. MT-9 cells were used for analyses of A3-recognizing epitopes.

### 4.2. Immunofluorescence of MT-9 Cells for A3

Cultured MT-9 cells at subconfulency were fixed by periodate-lysine-paraformaldehyde (PLP) fixative [40]. Chamber slide was permeabilized with 0.03% Triton X-100 in phosphate-buffered saline (PBS) after PLP-fixation and immunolabeled by A3 (1:1000: TransGenic, Inc., Kobe, Japan) for 1 h at RT. The visualization of specific antibody binding was performed with anti-mouse IgG secondary antibody conjugated with Alexa Fluor® 568 (Thermo Fisher Scientific Inc., Waltham, MA, USA). These chamber slides were coverslipped with Fluoro-KEEPER Antifade Reagent with 4′,6-diamino-2-phenylindole (DAPI) (Nacalai Tesque Inc., Kyoto, Japan) for nuclear staining. Slides were captured by VS120 Virtual Slide Microscope (Olympus Corporation, Tokyo, Japan) and analyzed using Olympus OlyVIA software (Olympus Corporation, Tokyo, Japan).

### 4.3. Isolation of A3 Antigen from Cultured MT-9 Cells

Lysates of cultured MT-9 cells was prepared as follows: the cells with confluents in Nunc EasYFlask 75 cm^2^ were stripped by a scraper; then, these cells were agitated by needle aspiration and incubation after adding CelLytic-MT mammalian cell lysis buffer (Sigma-Aldrich, St. Louis, MO, USA) for 30 min on ice. Supernatant was centrifuged (20,000× *g*, 30 min, 4 °C) to remove insoluble residues. The supernatant was mixed with an equal volume of 2× Tris-sodium dodecyl sulfate (SDS) sample buffer (Cosmo Bio Co., Ltd., Tokyo, Japan), which contained 0.125-M Tris-HCl, 4.3% SDS, 30% Glycerol and 0.01% bromophenol blue (BPB) (pH 6.8), with 5% 2-mercaptoethanol (BME) as a reducing solution or without BME. The mixtures for SDS-PAGE were incubated at room temperature, for 3 min, or boiled at 95 °C for 3 min; the final concentration was 2.0-mg/mL just before electrophoresis.

### 4.4. Western Blotting

Samples prepared from cultured MT-9 cells were developed by electrophoresis on 10% SDS-polyacrylamide gel (e-PAGEL(R); ATTO Corporation, Tokyo, Japan) under nonreducing conditions, followed by either Western blotting or protein staining. For Western blotting, the proteins were transferred to a polyvinylidene difluoride (PVDF) (Immun-Blot® PVDF Membrane; Bio-Rad Laboratories Inc., Tokyo, Japan), followed by immunoblot detection with A3 (0.1 μg/mL). The gel was directory applied to SDS-polyacrylamide gel (ATTO Corporation). For detection of the A3 antigen, Histofine® Simple Stain Rat MAX PO (M) (Nichirei Bioscience Inc., Tokyo, Japan) was used for a PVDF membrane with ECL Prime Western Blotting Detection Reagent (GE Healthcare Life Sciences, Buckinghamshire, UK) for chemiluminescence detection. The images of the A3 signal and gel staining were scanned by using GT-X800 (SEIKO Epson Corporation, Nagano, Japan). For chemiluminescence detection, the image was obtained by ImageQuant LAS 4000 (GE Healthcare Life Sciences, Buckinghamshire, UK) and analyzed using Fujifilm Multi Gauge Software (FUJIFILM Medical Co. Ltd., Tokyo, Japan).

### 4.5. Periodic Acid–Schiff (PAS) Stain

PVDF membrane, which was blotted with MT-9 lysate, was stained with PAS reagent to examine a possibility whether A3 recognizes a carbohydrate or not. The PVDF membrane was oxidized in 0.5% periodic acid solution for 10 min, followed by Schiff reagent for 60 min. After the staining, the PVDF membranes were washed by sulfurous acid solution for 9 min and then distilled water.

### 4.6. Glycosidase Analysis

To determine whether glycoepitope was an A3-recognizing antigen or not, reaction mixtures consisting of MT-9 lysate (25 μg) with peptide-N4-(N-acetyl-β-glucosaminyl)-asparagine amidase (Glycopeptidase F; Takara Bio Inc., Shiga, Japan) (10 mU) and control glycopeptide (25 μg) with glycopeptidase F (10 mU) were digested under nonreducing conditions: 37 °C, 19 h. The digests were applied to immunoblotting and developed with the A3 antibody.

### 4.7. Histological Analysis

#### 4.7.1. Animals

F344/DuCrj strain rats more than 5 weeks, and pregnant F344/DuCrj rats were obtained from Charles River, Japan (Shiga, Japan). Skin tissues were prepared for mature hair follicles from F344/DuCrj rats more than 6 weeks old; in addition, hair follicles with hair cycles on neonatal days 4 to 21 after birth were evaluated. Additionally, the bone marrow of Crl:CD (SD) rats at 4 weeks of age, which were obtained from Charles River, Japan, were evaluated.

For the wound-healing study, a cutaneous wound was made as follows: F344/DuCrj male rats at 10 weeks of age (Charles River, Japan) were used, and after shaving the dorsum, three circular (five-mm in diameter) full-thickness skin wounds were made with a sterile biopsy punch (Maruho Co., Ltd., Osaka, Japan) in the right and left of the back midline of the rats. Cutaneous wound tissues were harvested on post-wounding days 1, 3, 5, 7, 9, 12, 15, 20 and 26; rats without wounds served as controls and were sacrificed on PW day 0.

Animal housing and sampling conformed to the institutional guidelines of animal care of Osaka Prefecture University (Approval Nos. 28-1, 29-4, 30-2 and 19-51 during years 2016–2019).

#### 4.7.2. Tissue Preparations

Skin tissues were harvested from the trunk parallel to the vertebral line. Bone marrow was taken from the femur. All rats were euthanized by exsanguination under deep isoflurane anesthesia. Sample tissues were fixed in PLP fixative; PLP-fixed specimens were processed with the Acetone-Methyl benzoate-Xylene (AMeX) method [13]. Some tissues were embedded in embedding medium (TISSU MOUNT; Chiba Medical Co., Ltd., Saitama, Japan) and kept at −80 °C for frozen tissue sections.

#### 4.7.3. Immunohistochemistry and Double-Immunofluorescence

The tissue sections were used for immunolabeling with the primary antibodies, which are shown in Table 1. The bound antibodies were detected with horse radish peroxidase-conjugated anti-rabbit secondary antibody (Histofine Simple stain MAX-PO; Nichirei Bioscience Inc., Tokyo, Japan) and 3,3′-diaminobenzidine (DAB) substrate kit (Vector Laboratories, Burlingame, CA, USA) as chromogen. For double-immunofluorescence of A3 with other antibodies, sections were incubated with Alexa Fluor® 488-labeled or Alexa Fluor® 568-labeled secondary antibodies against mouse IgG (1:1000; Thermo Fisher Scientific Inc., Waltham, MA, USA) for A3 or were incubated with Alexa Fluor® 568-labeled secondary antibody against goat IgG (1:1000; Thermo Fisher Scientific Inc., Waltham, MA, USA) for CD34. These immunofluorescence sections were covered by a mounding medium with DAPI (VECTASHIELD®; Vector Laboratories Inc., Burlingame, CA, USA) for nuclear staining. Slides were scanned using VS120 Virtual Slide Microscope (Olympus Corporation, Tokyo, Japan) and analyzed using Olympus OlyVIA software (Olympus Corporation, Tokyo, Japan).

## 5. Conclusions

The present study showed that a newly developed antibody, A3, labeled somatic stem cells in rat bone marrow. Furthermore, it was found that A3 can label both epithelial and mesenchymal immature cells in the rat skin, contributing to hair follicle cycle and cutaneous regeneration (re-epithelialization) after wound. A3 would become a useful marker to identify somatic stem cells capable of differentiating both epithelial and mesenchymal cells in rat tissues.

## Figures and Tables

**Figure 1 ijms-21-03806-f001:**
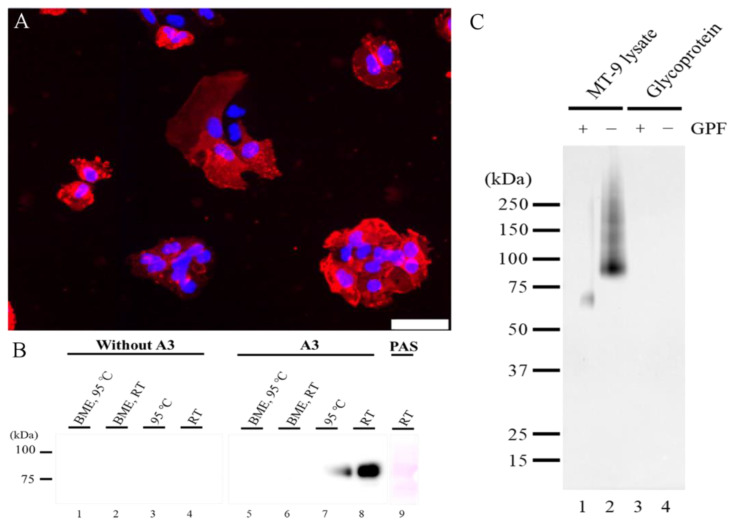
(**A**) A3 antigen in MT-9 cells. A3 antigen appears diffusely on the cell surface of MT-9 cells. Furthermore, fine granular reactions to A3 are also observed in the cytoplasm of MT-9 cells. Scale bar = 50 μm. (**B**) A3 reactivity. In Western blotting without the primary antibody, A3 does not show any signals in lanes 1–4. Samples treated with a reducing reagent (BME) lose their antigenicity in lanes 5 and 6. Specific A3-signal band is observed at 75–100 kDa in lanes 7 and 8. The strongest A3-signal band is observed in a sample incubated at RT without BME in lane 8. A weaker signal is also observed in a sample incubated at 95 °C without BME in lane 7. PAS, periodic acid-Schiff; BME, 2-mercaptoethanol and RT, room temperature. (**C**) Glycodigestion. A3 antigen digested with glycopeptidase F (Lane 1) and intact A3 antigen (Lane 2) were analyzed by Western blotting for A3. For the negative control, control glycoprotein with glycopeptidase F (Lane 3) and intact glycoprotein without glycopeptidase F (Lane 4) were analyzed by Western blotting for A3. GPF, glycopeptidase F.

**Figure 2 ijms-21-03806-f002:**
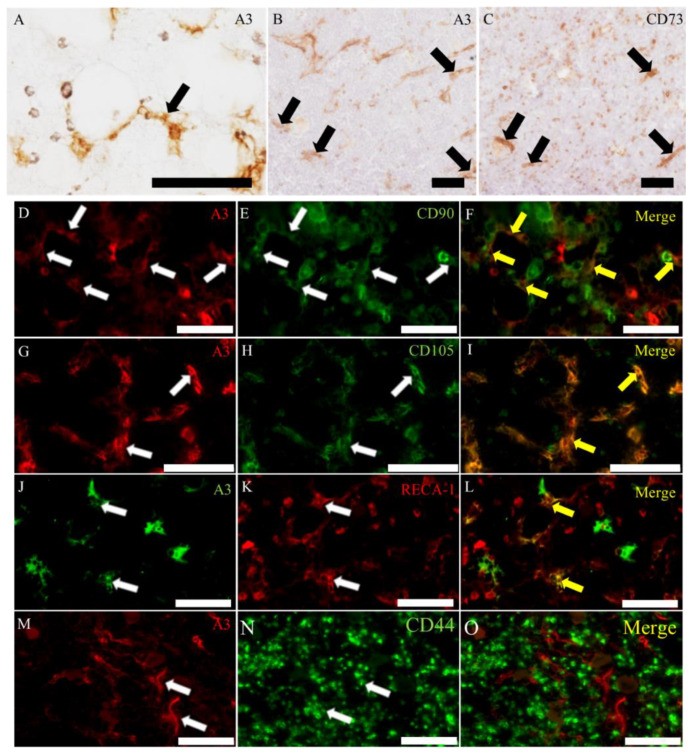
A3-labeled cells in the adult rat bone marrow. (**A**) A3-labeled cells are spindle- and polyhedral-shaped stromal cells. (**B**,**C**) The localization is similar partly to CD73-positive cells in the consecutive sections. (**D**–**L**) A3-labeled cells are corresponding to some cells reacting to CD90 (**D**–**F**), CD105 (**G**–**I**) or RECA-1 (**J**–**L**). (**M**–**O**) The localization of A3-labeled cells is different from that of CD44-positive cells. Black and white arrows indicate positive signals. Yellow arrows indicate colocalization in merge immunofluorescence. Scale bars = 50 μm.

**Figure 3 ijms-21-03806-f003:**
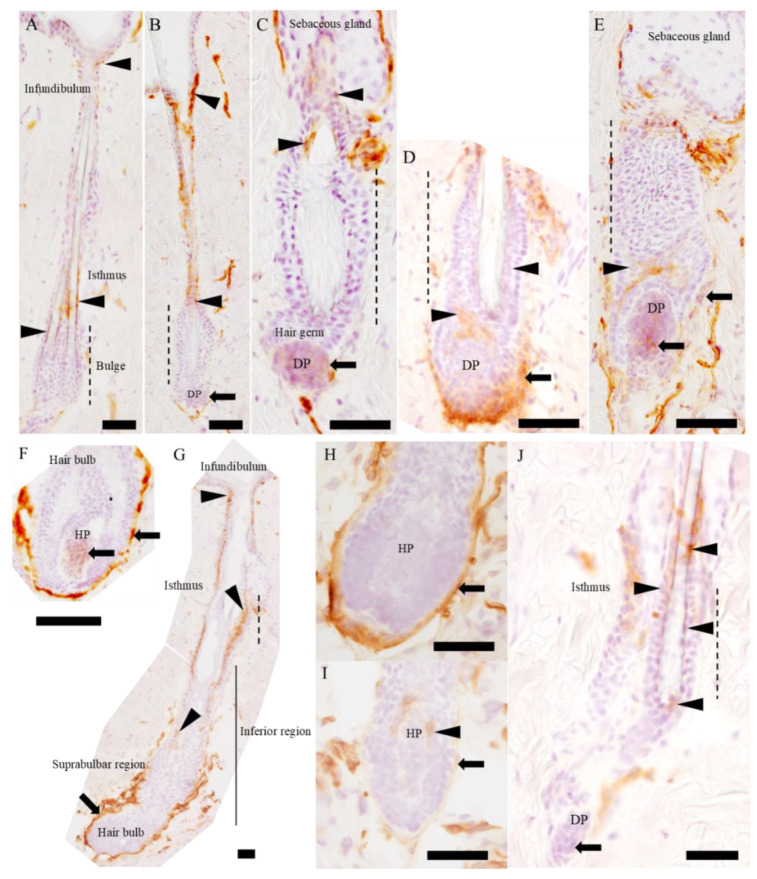
Localization of A3-labeled cells in the hair cycle. (**A**) A3-labeled cells are observed in the infundibulum and lower isthmus in the permanent region. (**B**) At the beginning of anagen, the A3-expressing area is gradually spreading, and A3 labels the mesenchymal cells in dermal papilla with weak reactivity. (**C**–**E**) High magnification of the bulge and dermal papilla. Hair follicle grows down into the dermis with the forming connective tissue sheath. (**F**) A3 labels the regenerating hair papilla, as well as the connective tissue sheath around the hair bulb. (**G**) A3 labels suprabasal cells in the hair follicle leading out of the bulge and mesenchymal cells in the connective tissue sheath with strong reactivity. A3 does not react to the mature hair papilla. (**H**,**I**) A3 does not react to the regressing hair papilla; the intensity of the A3 signal in the connective tissue sheath is gradually declined; A3 labels the epithelial cells on the hair papilla in the middle stage of catagen. (**J**) In the final stage of catagen, A3-labeled cells are observed in the bulge, infundibulum, isthmus and dermal papilla. Arrowheads indicate A3-labeled epithelial cells, and arrows indicate A3-labeled mesenchymal cells. DP, dermal papilla and HP, hair papilla. Dashed lines in (**A**–**E**,**G**,**J**) indicate the bulge region. Scale bars = 50 μm.

**Figure 4 ijms-21-03806-f004:**
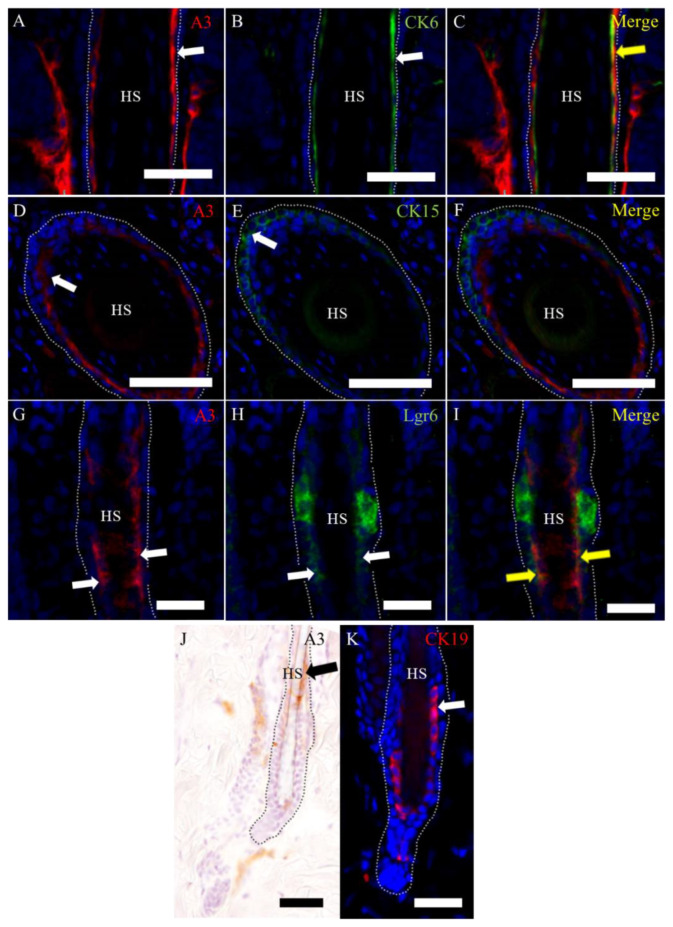
Characteristics of A3-labeled epithelial cells in the hair cycle. (**A**–**C**) A3 labels epithelial cells under the companion layer expressing CK6. (**D**–**F**) A3 labels epithelial cells above the basal layer expressing CK15. A3-labeled cells show partly colocalization with that for Lgr6. (**G**–**I**) The distribution of positive cells to A3 and CK19 (**J**,**K**; **J** for A3 and **K** for CK19) is almost similar to each other. Black and white arrows indicate positive signals. Yellow arrows indicate colocalization in merge immunofluorescence. HS, hair shaft. Dot lines indicate the basal membrane. Scale bars = 50 μm.

**Figure 5 ijms-21-03806-f005:**
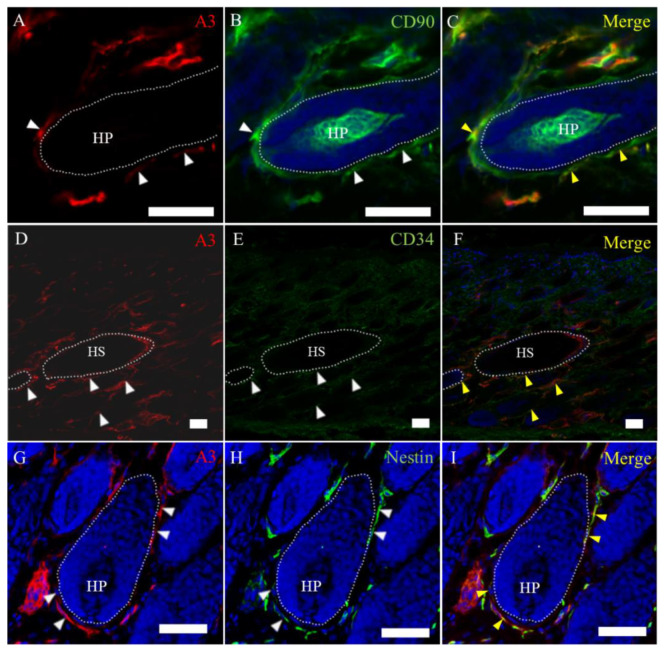
Characteristics of A3-labeled mesenchymal cells in the hair cycle. (**A**–**C**) A3 reacts to cells showing positive reaction to CD90 around the hair bulb. (**D**–**F**) CD34-positive cells are diffuse in the dermis; some of the CD34-positive cells react to A3. (**G**–**I**) A few cells show coexpression to A3 or nestin around the hair bulb. White arrowheads indicate positive signals. Yellow arrowheads indicate colocalization. HP, hair papilla and HS, hair shaft. Dot lines indicate the basal membrane. Scale bars = 50 μm.

**Figure 6 ijms-21-03806-f006:**
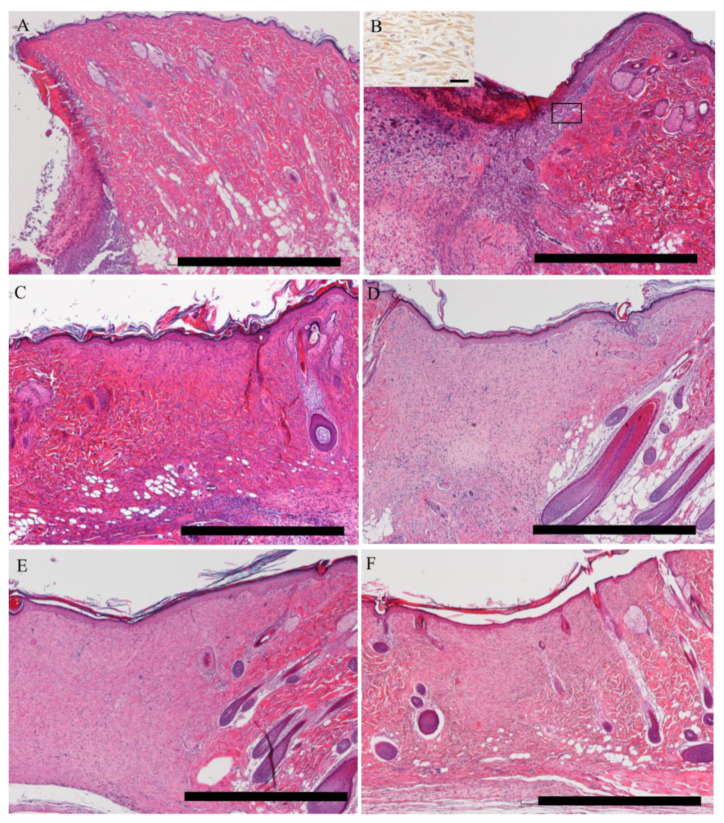
The wound-healing process. (**A**) In the inflammatory stage, the wound is covered by debris/degenerative cells and reactive cells such as neutrophils, fibrin and hemolyzed red blood cells, forming a crust. The epidermal layer is thin at the edge of the wound, and the hair follicles at the wound edge show the catagen phase on post wound (PW) day 1. (**B**) Thickened epidermis is formed from the surrounding cutaneous tissue; in the wound bed, granulation tissue with myofibroblasts (square is the inset showing the increased number of myofibroblasts reacting to α-SMA; scale bar = 20 μm) develops. (**C**,**D**) In the late-proliferative and early remodeling stages, the hair follicle at the wound edge shows anagen, and the thickened re-epithelialized epidermis is clear. (**E**,**F**) In the remodeling stage, the hair follicle shows catagen, and the epidermal thickening declines to the control level. A, PW day 1; B, PW day 5; C, PW day 9; D, PW day 15; E, PW day 20 and F, PW day 26. Scale bars = 1 mm.

**Figure 7 ijms-21-03806-f007:**
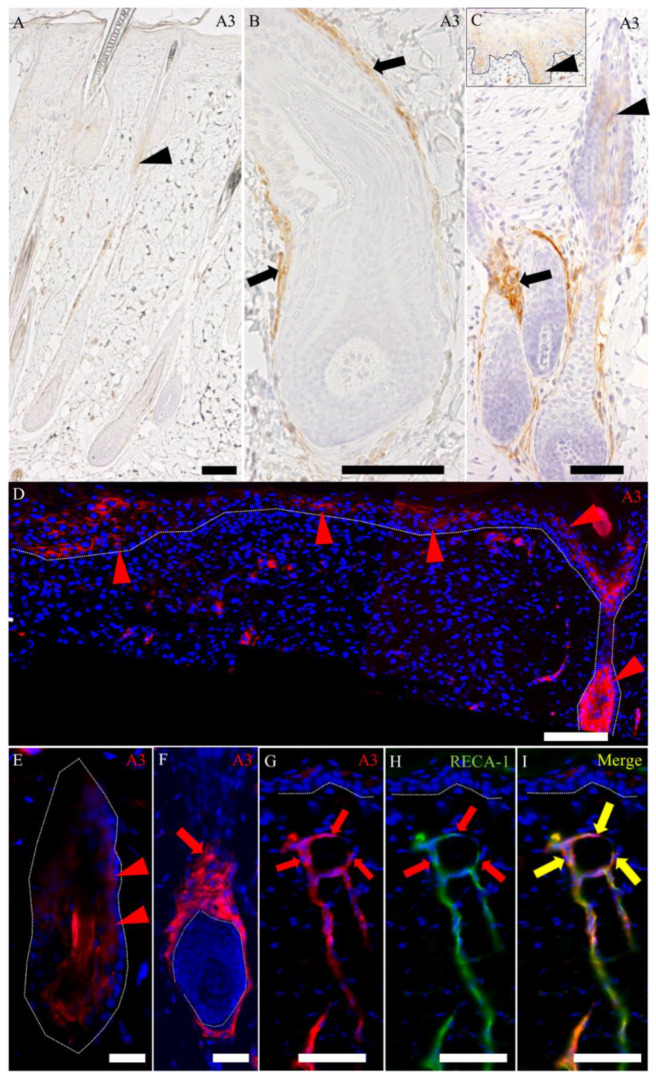
Distribution of A3-labeled cells in the cutaneous wound healing. (**A**) In the control skin, A3 labels both the epithelial and mesenchymal components of the hair follicles, although A3-labeled mesenchymal cells in the connective tissue sheath are rarely seen. (**B**–**D**) In the late-proliferative stage, A3 reacts to both the epithelial and mesenchymal components of the anagen hair follicle at the edge of the wound bed, showing the greater signal intensity. A3 also reacts to re-epithelialized epidermis, which are continuous from the hair follicle at the edge of wound bed. Inset in (**C**) shows epidermal invaginations in re-epithelialized epidermis, and (**D**) shows the wound bed and hair follicle at the wound edge. (**E**,**F**) Signal intensity to A3 in follicular epithelial (**E**) and mesenchymal (**F**) cells declines in the late-remodeling stage, compared to that in the late-proliferative and early remodeling stages, although the intensity is still greater than that in the control skin. (**G**–**I**) A3 also reacts to pericytes in the wound bed. Black and red arrows indicate positive signals for the mesenchymal cells. Black and white arrowheads indicate positive signals for the epithelial cells. Yellow arrowheads indicate colocalization. Dot lines in C, D, E, F, G, H and I indicate the basal membrane. Scale bars (**A**–**D**) = 100 μm. Scale bars (**E**–**I**) = 50 μm.

**Figure 8 ijms-21-03806-f008:**
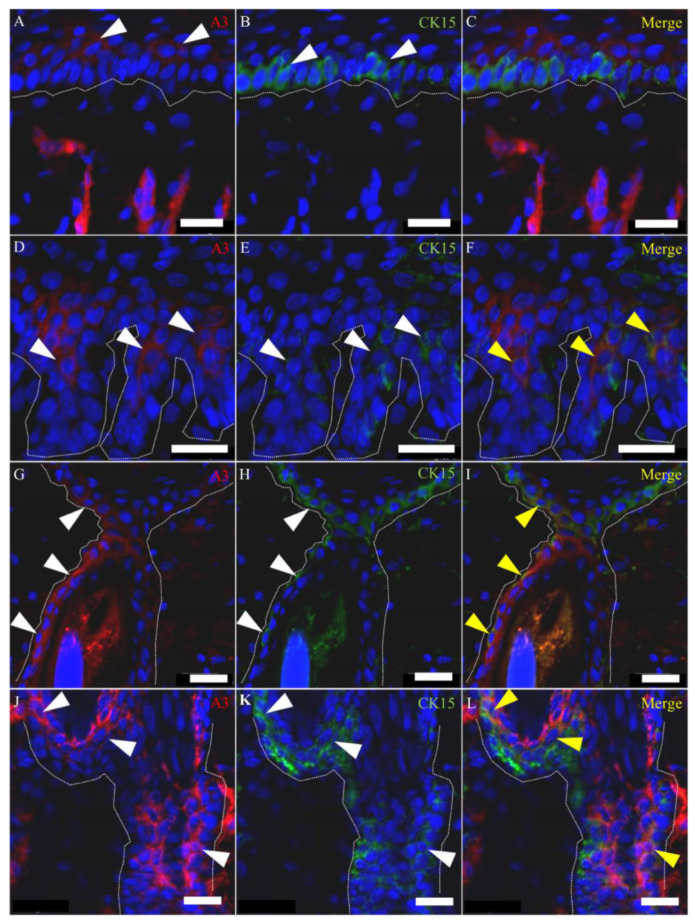
Double immunolabeling with A3 and CK15 antibody in the early remodeling stage. (**A**–**C**) A3 labels suprabasal cells, which are contiguous to CK15-positive cells in the re-epithelialized epidermis. (**D**–**F**) In epidermal invaginations, CK15 reacts not only to basal cells but also to suprabasal cells, which are labeled by A3. (**G**–**L**) The colocalization of A3 with CK15 is also observed at the infundibulum (**G**–**I**) and bulge region (**J**–**L**) in the hair follicle at the edge of the wound bed. (**A**–**L)** Samples on PW day 15. White arrowheads indicate positive signals. Yellow arrowheads indicate colocalization. Dot lines indicate the basal membrane. Scale bars = 20 μm.

**Figure 9 ijms-21-03806-f009:**
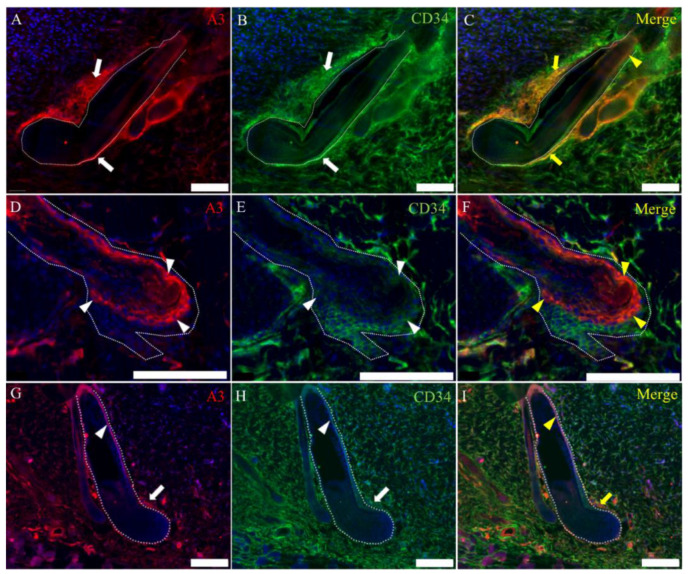
Double immunolabeling with A3 and CD34 antibody. (**A**–**C**) CD34 antibody labels immature mesenchymal cells in the dermis, and the antibody does not react to mesenchymal cells in the granulation tissue in the early remodeling stage at the left-upper side in the figures. A3 reacts to some CD34-positive mesenchymal cells in the connective tissue of the hair follicle at the edge of the wound bed in the early remodeling stage. (**D**–**F**) CD34 antibody labels the bulge epithelial cells and some epithelial cells reacting to A3 in the early remodeling stage. The area of A3-labeled cells is narrower than that of the CD34-positive cells. (**G**–**I**) The connective tissue sheath is thinned in the late-remodeling stage on PW day 26, compared with that in the early remodeling stage on PW day 15. (**A**–**F**) Samples on PW day 15 and (**G**–**I**) samples on PW day 26. White arrowheads and arrows indicate positive signals for epithelial and mesenchymal cells, respectively. Yellow arrowheads indicate epithelial colocalization, and arrows indicate mesenchymal colocalization. Dot lines indicate the basal membrane. Scale bars = 100 μm.

**Figure 10 ijms-21-03806-f010:**
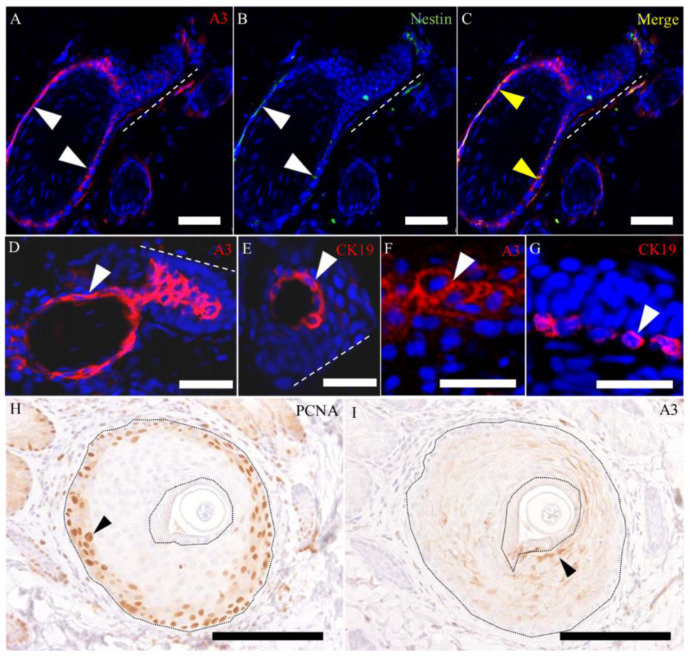
The characteristics of A3-labeled epithelial cells in the late-proliferative and early remodeling stages. (**A**–**C**) A3 reacts to nestin-positive epithelial cells around the bulge. (**D**–**G**) The A3- and CK19-labeled epithelial cells (**D**,**F**) are seen around the bulge (**E**) in the figures of D and F and in the re-epithelialized epidermis (**G**) in the figures of F and G. (**H**,**I**) Proliferating cell nuclear antigen (PCNA)-positive proliferating cells are located in the outer few layers in the outer root sheath, although A3 reacts to the inner cells of the outer root sheath, except for the basal cells. White arrowheads indicate positive signals in the epithelial elements. Yellow arrowheads indicate colocalization. (**A**–**G)** Samples on PW day 15 and (**H**,**I**) samples on PW day 9. Dashed lines in A–D and E indicate the bulge region, and dot lines in H and I indicate the outer root sheaths. Scale bars (**A**–**C**,**H**,**I**) = 40 μm. Scale bars (**D**–**G**) = 20 μm.

**Figure 11 ijms-21-03806-f011:**
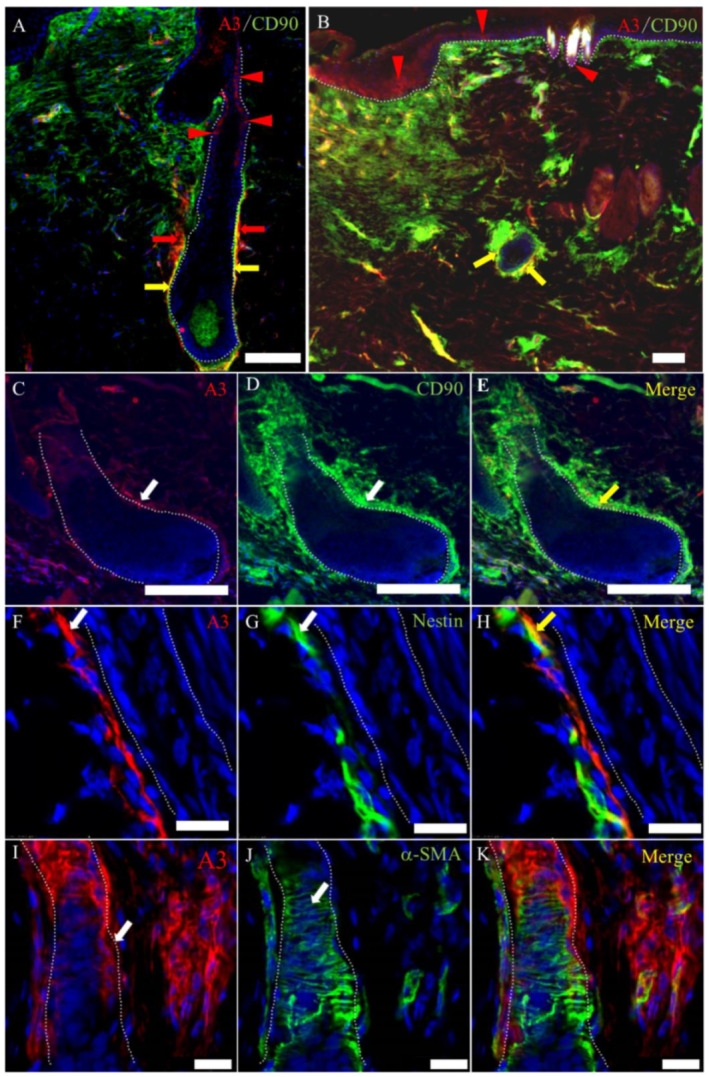
The characteristics of A3-labeled mesenchymal cells in the late-proliferative and early remodeling stages. (**A**–**E**) A3 reacts to the expanded immature mesenchymal cells, which coexpress CD90 in the connective tissue sheath of the hair follicle in the proliferative and remodeling stages. CD90 also reacts to the mesenchymal cells in the granulation tissue, especially beneath the re-epithelialized epidermis. (**F**–**H**) A3 reacts to some nestin-positive cells. (**I**–**K**) A3-labeled cells are around the hair follicle. α-SMA-positive cells are arranged in parallel around the hair follicle. (**A**,**B**) Samples on PW day 9, (**C**–**H**) samples on PW day 15 and (**I**–**K**) samples on PW day 12. Red arrowheads and arrows indicate the A3-labeled epithelial and mesenchymal cells, respectively, in (**A**,**B**). White arrows indicate positive signals for mesenchymal cells in (**C**,**D**,**F**,**G**,**I**,**J**). Yellow arrowheads and arrows indicate colocalization in A, B, E, and H. Dot lines indicate the basal membrane. Scale bars (**A**–**E**) = 100 μm. Scale bars (**F**–**K**) = 20 μm.

**Table 1 ijms-21-03806-t001:** Primary antibody list.

PrimaryAntibody	Clone	Poly/Mono	Dilution	Source of Antibody	Specificity
A3	A3	Mouse mono	1000	TransGenic Inc.,Hyogo, JPN	-
α-SMA	1A4	Mouse mono	500	Dako, Carpinteria, CA, USA	Smooch musclecells, myofibroblasts
Cytokeratin 6	LHK6B	Mouse mono	200	Thermo Fisher Scientific Inc.,Waltham, MA, USA	Epithelial cell, inner root sheathcells
Cytokeratin 15	LHK15	Mouse mono	200	Thermo Fisher Scientific Inc.,Waltham, MA, USA	Follicular basalcells (stem cells) inthe bulge
Cytokeratin 19	B170	Mouse mono	200	Leica Biosystems, Eisfeld, GER	Follicular stem cell
CD34	-	Goat poly	200	R&D Systems, Minneapolis, MN, USA	Stem cells
CD44 FITC	OX49	Mouse mono	500	BD Pharmingen Inc. San Jose, CA, USA	Stem cells
CD73	5F/B9	Mouse mono	500	BD Pharmingen Inc. San Jose, CA, USA	Immaturemesenchymal cells
CD90 Alexa Fluor 488	OX-7	Mouse mono	500	Bio-Rad Laboratories Inc., Hercules, CA, USA	Mesenchymal stemcells
CD105 Alexa Fluor 488	SN6	Mouse mono	500	Bio-Rad Laboratories Inc., Hercules, CA, USA	Mesenchymal stemcells
Lgr6	EPR6874	Rabbit mono	200	Abcam, Cambridge, UK	Stem cell in theisthmus
Nestin	25	Mouse mono	500	BD Pharmingen Inc. San Jose, CA, USA	Stem cells
RECA-1 labeled by Alexa Fluor 555 Mouse IgG_1_ labeling kit	RECA-1	Mouse mono	200	Abcam, Cambridge, UK	Endothelial cells
PCNA	PC-10	Mouse mono	50,000	Dako, Carpinteria, CA, USA	Proliferating cells

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
