# Peer review of "Participation of Somatic Stem Cells, Labeled by a Unique Antibody (A3) Recognizing Both N-glycan and Peptide, to Hair Follicle Cycle and Cutaneous Wound Healing in Rats"

_ijms, 2020, doi:10.3390/ijms21113806_

Round 1

Reviewer 1 Report

All my concerns were addressed by the authors. 

Reviewer 2 Report

no comments

Reviewer 3 Report

The authors have addressed all questions from previous review. The manuscript can be published with current version. 

This manuscript is a resubmission of an earlier submission. The following is a list of the peer review reports and author responses from that submission.

Round 1

Reviewer 1 Report

In this manuscript, Chisa et. al. reported that A3-labeled cells co-express immature mesenchymal markers and participate in hair cycle and wound healing, indicating A3 could be a candidate to identify somatic stem cells involved in both epithelial and mesenchymal cells differentiation in rat. The study is interesting, and the results can be valuable for understanding somatic stem cell biology. However, some experimental designs contain flaws, and the data presentations need to be improved.  

  1. The current title does not fit well with the content and needs to be adjusted. The title states ‘recognizing both N-glycan and peptide’, however, this paper only describes the participation of A3-labeled cells during hair cycle and wound healing.
  2. The first three figures should be combined, because they are supporting the same point. In line 69, explain MT-9 in the figure/context.
  3. Fig2 & 3 are lacking of internal controls.
  4. Fig3, in MT-9 lysate without GPF treatment group, explain the smearing bands.
  5. Fig4B and C cannot support the conclusion in line 102. Immunofluorescence images should be provided.
  6. Fig4, CD90, CD105 and CD44 are all mesenchymal stem cell markers. Explain why A3-labeled cells only co-expressed with CD90 and CD105, not CD44.
  7. Fig4J and K, double check to see if these two figures were switched/mislabelled.
  8. Fig5, 8, and 9, cartoon diagrams could be provided to show the transition of A3-labeled cells during hair cycle and wound healing process.
  9. Fig6G-I, it seems that Lgr6 and A3 are not co-localized with each other in the images.
  10. Fig6J-K, either IHC staining of CK19 or immunofluorescence staining of A3 should be included.
  11. Fig8B, explain the inserted small panel on the top left, and include a scale bar in it.
  12. Fig12D-G, double immunolabelling images for A3 and CK19 should be added.
  13. More details relevant to the carbohydrates binding to A3 antibody can be discussed. The function of A3-labeled cells in hair cycle (line 366) or wound healing (line 416) and how the discovery is going to guide stem cell biology research should be further elaborated.

Reviewer 2 Report

The manuscript by Katou-Ichikawa and co-authors describes the analysis of molecular biological features of the epitope recognized by the antibody A3 and focuses on the capability of A3 to mark somatic stem cells in hair follicle cycle and cutaneous wound repair in rats.

The authors presented many experiments showing the aim of the work, but the manuscript needs to be strongly revised.

Major points:

-The authors have to reorganize the results section. All experiments must be better described and mostly the rationale of each experiment must be clarified.

The experiments in fig.2 lack of reference protein (e.g. beta-actin), moreover the PAS staining must be improved.

The localization of A3 and other markers must be indicated in all figures by an arrow.

-The discussion must be improved discussing the biological relevance of this A3 antibody.

Reviewer 3 Report

This manuscript by Katou-Ichikawa C et al. described the localization of somatic stem cells labeled by A3 antibody during hair follicle cycle and wound healing processes. The authors characterized the antigen of A3 antibody by western blotting and GPF treatment. Immunohistochemistry and immunofluorescence analysis showed the localization of A3-positive cells in the hair follicles and skin wound healing. By co-staining with other stem cell markers, the authors analyzed the characteristics of A3 positive cells in both epithelial and mesenchymal regions. The experiments were well-performed, and the manuscript was clearly written; however, there are several concerns as shown below.

Comments:

  1. In the title, the authors used the term “a novel antibody”; however, the authors published about A3 antibody in several articles. Thus, “a novel antibody” should not be used in the title.
  2. In Figure 4, A3-positive cells were almost overlapped with CD105-positive cells, which is a typical MSC marker. But, A3 and CD44 were not overlapped at all; the authors mentioned that CD44 is also a MSC marker. How the authors explain this discrepancy. Does A3 recognize MSC?
  3. To separate epithelia and mesenchyme, dot lines are used in some figures (such as Fig. 9-11). This is convenient to see the figures. The authors should add the dot lines in other figures such as Figures 6, 7, and 13.